# Effects of Low-Severity Fire on Soil Physico-Chemical Properties in an Andean Páramo in Southern Ecuador

Vinicio Carrión-Paladines [1],*, Andreas Fries [2], María Belén Hinojosa [3], Andy Oña [1], Leticia Jiménez Álvarez [1], Ángel Benítez [1], Fausto López Rodríguez [1] and Roberto García-Ruiz [4]

1. Departamento de Ciencias Biológicas, Universidad Técnica Particular de Loja, San Cayetano Alto s/n C.P., Loja 11-01-608, Ecuador; ajona1@utpl.edu.ec (A.O.); lsjimenez@utpl.edu.ec (L.J.Á.); arbenitez@utpl.edu.ec (Á.B.); fvlopezx@utpl.edu.ec (F.L.R.)
2. Departamento de Geología, Minas e Ingeniería Civil (DGMIC), Universidad Técnica Particular de Loja, San Cayetano Alto s/n, Loja 11-01-608, Ecuador; aefries@utpl.edu.ec
3. Departamento de Ciencias Ambientales, Universidad de Castilla-La Mancha, Campus Fábricas de Armas, 45071 Toledo, Spain; mariabelen.hinojosa@uclm.es
4. Departamento de Biología Animal, Biología Vegetal y Ecología, Sección de Ecología, Universidad de Jaén, Campus Las Lagunillas, 23071 Jaén, Spain; rgarcia@ujaen.es
* Correspondence: hvcarrionx@utpl.edu.ec; Tel.: +593-967907558

**Abstract:** The high Andean páramos (AnP) are unique ecosystems that harbor high biodiversity and provide important ecosystem services, such as water supply and regulation, as well as carbon sequestration. In southern Ecuador, this ecosystem is threatened by anthropogenic burning activities to create pastures and agricultural land. However, knowledge of the effects of fire on soil properties and nutrient availability is still limited. This study conducted an experimental burn with different ignition patterns on an AnP plateau in southern Ecuador. Fire behavior (flame height, propagation speed, temperature reached on the soil), and fire severity were evaluated. In addition, soil samples were collected at 10 cm depth both 24 h and one year after the burns to measure the effects of fire on the main physico-chemical properties. The results indicate that the low severity of the experimental burns did not affect the physico-chemical properties of the soils, and therefore, soil quality was not altered. These results can help decision makers in the design of policies, regulations, and proposals for the conservation and environmental restoration of AnPs affected by wildfires in southern Ecuador.

**Keywords:** Andean páramo; fire ecology; low-severity fire; soil nutrient dynamics





## 1. Introduction

Andean páramos (AnPs) are one of the world's most rapidly evolving biodiversity hotspots [1]. These ecosystems are found at altitudes ranging from 3000 to 3800 m above sea level and are characterized by treeless vegetation and a diversity of lakes and peatlands [2]. AnPs are found in the South American Andean Mountain ranges, including Colombia (14,434 km$^2$), Venezuela (2660 km$^2$), Ecuador (12,600 km$^2$), and northern Peru (4200 km$^2$) [3]. They are also found in small areas of the Cordillera de Talamanca in Costa Rica and Panama (80 km$^2$) [4,5]. Due to their altitudinal and topographic positioning, the AnPs are adapted to a wide range of environmental conditions [3]. AnPs are key to the capture, regulation, and supply of water for nearby areas [6]. This is due to the hydromorphic characteristics of the soil that allow them to retain water, as they develop in water-saturated conditions due to the constant precipitation and relatively low evapotranspiration of this ecosystem throughout the year [7,8]. In addition, these soils are characterized by low bulk density, high porosity, and very friable consistency, which limits their use for agriculture activities and makes them vulnerable to soil compaction by continuous trampling of livestock [9]. According to their physical characteristics, páramo soils have many properties in common with peatlands [2,6]. They can contain up to 44% organic matter, which can reach up to 90% in peatlands [10].

The climate at these high elevations is cold and humid, with sudden fog and drizzle, and rapid changes in temperature, solar radiation, and humidity during a day [2,11]. The cold and humid climate favors the accumulation of organic matter in the soil, which plays an important role in carbon storage.It explains its high porosity and microporosity compared to other soil types [12,13], and it is recognizable by its dark humic soil layer [14]. This accumulation of organic matter and carbon lead to a high water regulation capacity of the AnP soils, which is also facilitated by the topography containing many local concavities and depressions forming peatlands (wetlands) and small lakes [9]. The high water retention capacity of these soils helps to reduce downstream flood risks and to ensure a constant river flow for domestic, industrial, and agricultural uses throughout the year [15,16].

AnPs are also fragile ecosystems that face threats from human activities such as conversion to pasture for livestock production [17,18]. These ecosystems are home to many unique plant species; however, a significant portion of them are lost due to environmental changes brought on by rural development [19]. According to Buytaert et al. [9], AnPs house approximately 5000 plant species, with 60% being endemic, showcasing their exceptional biological diversity [20]. The native species are highly adapted to the prevailing edaphic and climatic conditions, often causing irreversible impacts to the ecosystems [21,22].

In Ecuador, AnPs occupy about 7% of the country's territory [23]. There are six different types of AnPs in Ecuador. In the northern and central regions, there are the herbaceous, the dry, the espeletias (with thick, hairy leaves that allow them to adapt to extreme conditions), and the cushion-forming plant (plants in the form of compact cushions that protect themselves from the cold and wind and retain water) páramos. In the southern region, there are herbaceous and shrubby páramos [24]. Despite their limited geographical extent, AnPs face significant anthropogenic pressures, including burning practices to create pasture and agricultural land [25]. These burning practices have historical roots [26,27], but the decline in traditional techniques has led to wildfires spreading more frequently to other ecosystems [28]. However, despite research regarding the hydrological capacity of soils [29,30], fire ecology [31], and traditional burning practices [32], knowledge about the effects of fire on soil properties and nutrient availability in AnPs is still limited, and this is because the behavior of fire in this ecosystem is unknown. Moreover, fire intensity and severity depend on vegetation type, fire climatology, soil type, and soil moisture content at the time of burning, which also determine ash production and quality [33,34].

In general, high-, medium-, and low-severity fires can occur when vegetation is burned. Of these, low-severity fires can have positive effects by accumulating organic matter in the soil due to the incomplete combustion of biomass [35]. However, the ashes produced could also have impacts on soil properties, such as affecting bulk density, as they can clog soil pores, reducing its water regulation capacity [33,36].

The objective of this study was to evaluate the effect of low-severity fire on the physico-chemical soil properties in an AnP in southern Ecuador through an experimental burn. For this purpose, a meteorological station (San Lucas_UTPL) was installed and permanent sampling plots were established. In addition, different ignition patterns and techniques were applied to produce different fire severities in the AnP and to analyze their effects on physico-chemical soil properties.

## 2. Materials and Methods

### 2.1. Study Area

The experimental burn area is located within the natural AnP of southern Ecuador, where the Saraguro indigenous parish resides (latitude 3°43′38.20″ S; longitude 79°12′30.25″ W; mean elevation 3050 m asl) (Figure 1a). The ecosystem can be classified as a humid montane forest with the presence of herbaceous páramo at higher elevations (Figure 1b) [24]. The soils are identified as Umbric Ferralsol, characterized by a surface layer rich in organic matter, an acidic pH, and a high iron and aluminum content [37,38]. The páramo has a regular topography due to its position within a high mountain plateau, as with other AnPs

in Central and South America [39]. This AnP is part of the northwestern buffer zone of the protected area "Corazón de Oro", which is managed by the local Saraguro community.

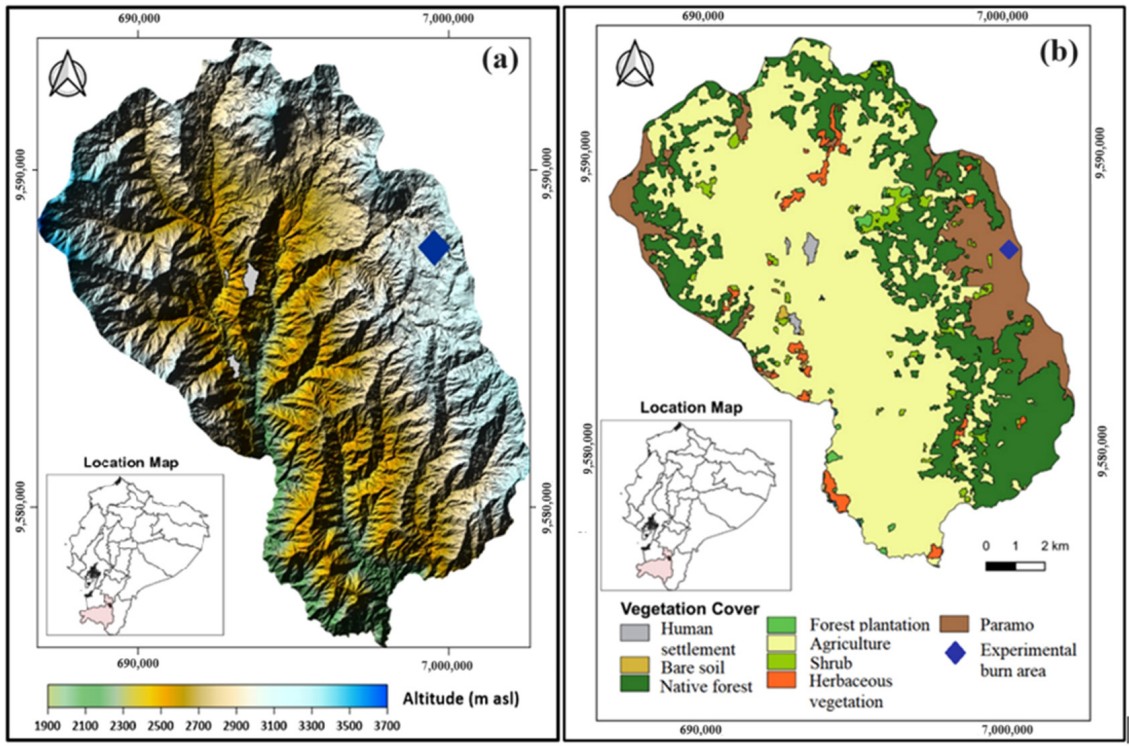

**Figure 1.** Digital elevation model (**a**,**b**) and principal land uses in the San Lucas parish in southern Ecuador. The blue rhombus indicates the experimental burning site.

The vegetation of this herbaceous páramo is composed of a variety of plants and grasses [6]. The main plant species include páramo straw (*Calamagrostis effusa*), fleshy-rooted plants such as *Eryngium humile*, *Xenophyllum humile*, and *Paepalanthus* spp., achupal-las rosettes (*Espeletia* sp.), blackberries (*Rubus ulmifolius*), miconias (*Miconia calvescens*), puya (*Puya hamata* L.B.Sm.), páramo straw (*Calamagrostis intermedia* [J.Presl] Steud.), páramo flower (*Oritrophium crocifolium* [Kunth] Cuatrec.), naurapo (*Myrteola nummularia* [Poir.] O. Berg.), sirius (*Xyris subulata* Ruiz & Pav.), and joyapas (*Disterigma rimbachii*). These plants serve as food sources for a variety of mammals such as the spectacled bear (*Tremarctos ornatus*) [40,41].

The herbaceous páramo provides crucial ecosystem services, with the most significant benefit being the water supply for the local population [32]. The main settlement in this region, San Lucas, is located at an altitude of 2800 m above sea level and has a temperate climate with average temperatures around 13.5 °C and an annual rainfall range of 600 to 1000 mm [41]. The primary sources of livelihood for the local population are agriculture, livestock, forestry, and artisan products [42]. Agricultural production mainly consists of crops of corn (*Zea mays*), beans (*Phaseolus vulgaris*), broad beans (*Vicia faba* L.), potatoes (*Solanum tuberosum* L.), and peas (*Pisum sativum* L.), grown for self-consumption or the local market, as well as medicinal plants and flowers [41]. The continued expansion of agricultural land and livestock production are the main environmental challenges in the area due to the land use changes through slash-and-burn activities [41,42]. Moreover, the páramo vegetation is frequently burned to regenerate the grass vegetation, which serves as a food source for domestic animals, particularly for cattle (*Bos taurus*) [32].

### 2.2. Experimental Design

The experimental burning site is located on a high mountain plateau with slopes ranging from 0 to 20%. We established nine experimental plots in an area of the herbaceous

páramo facing southeast with a slope of 15% (Figure 2). The selection of this site was based on the knowledge of the Saraguros indigenous people, who have vast experience in fire management in this area, which also allowed us to determine the oldest area in which a fire occurred and therefore the area of greatest fuel accumulation [32].

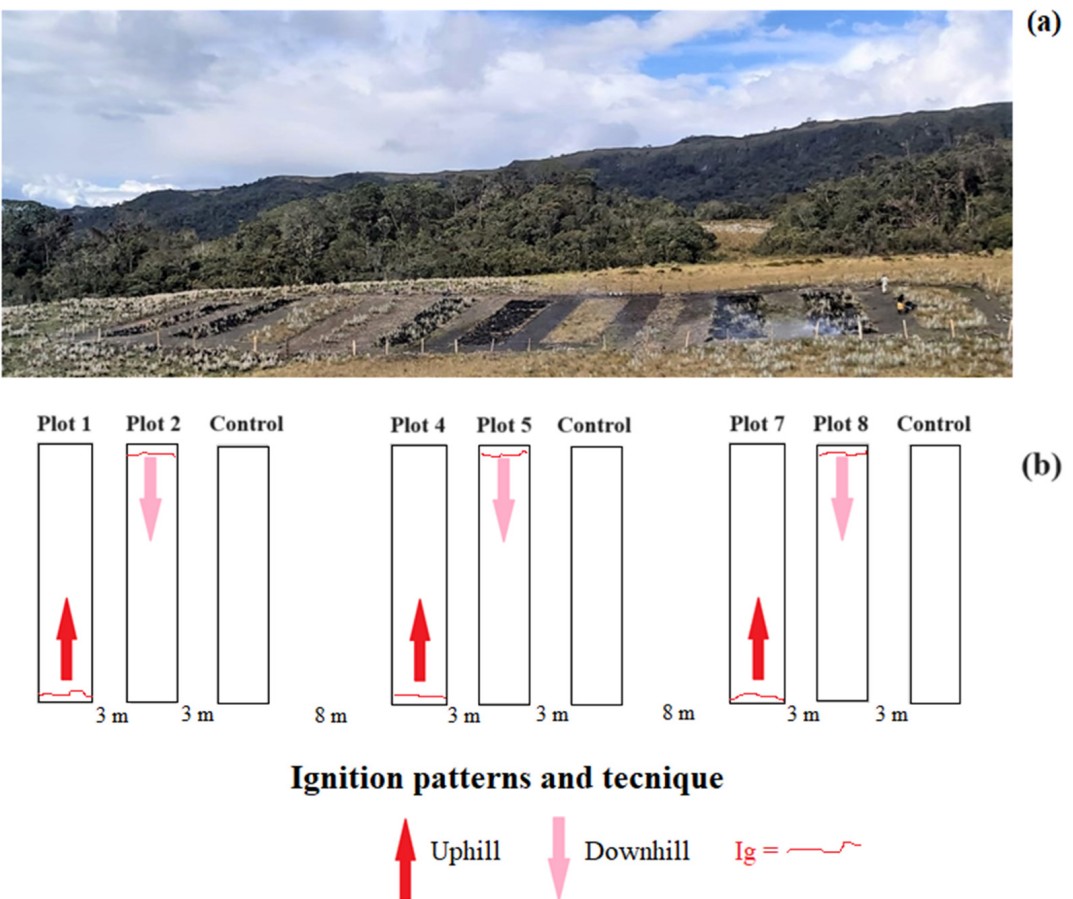

**Figure 2.** Experimental burning design in the study area: (**a**) photograph of the experimental site after burning, and (**b**) the burning pattern design, including the burning direction (upslope or downslope) and ignition starting line (Ig).

Individual plots were 4 m wide and 20 m long, with an area of 80 m$^2$. The individual plots were spaced 3 m apart, or 8 m, according to the scheme shown in Figure 2, to prevent fire spread. The plant material that was collected during plot delimitation was used as additional fuel, which was evenly distributed in the plots intended for controlled burning. This addition was justified by the fact that this technique reflects the traditional burning practices used in the region. In these practices, the Saraguro indigenous people build mounds with dried grass, typical of the herbaceous vegetation of the páramo, which are later incinerated [32]. Therefore, no additional material was incorporated in the control plots, as has been reported in previous studies [43]. The plots were protected with wooden posts and barbed wire to prevent access by cattle (*Bos taurus*) that occasionally graze in the area and native species such as the spectacled bear (*Tremarctos ornatus*) [41] (Figure 1). The main purpose was to protect natural regeneration, avoiding browsing and trampling in the experimental plots (Figure 2a).

The first plots of each block were burned from the bottom to the top (uphill) to create a frontal fire behavior (higher fire intensity and speed), while the second plots of each block were burned from the top to the bottom (downhill) to produce a backing fire behavior (lower fire intensity and speed) [44–46]. In addition, the third plot was kept unburned as a control plot (Figure 2a,b). A strip burn technique was applied for ignition (Ig), with

the starting line set 20 cm inside the plot to prevent fire spreading into the cut areas [47]. The ignition was induced using a mixture of diesel (3/4) and gasoline (1/4) in a 5 L fuel container. All plots were ignited using a single starting line (Ig, Figure 2b).

### 2.3. Evaluation of Fire Behavior and Severity during Experimental Burns

The experimental burning was conducted in 2021 during the Veranillo del Niño phenomenon (VdN), a period of dry and sunny weather, which typically occurs for approximately 15 days between October and November [32,48]. Meteorological conditions were monitored using a Vantage Pro Plus automatic weather station from Davis Instruments USA (San Lucas_UTPL), located close to the experimental site [49]. The monitored meteorological variables included temperature (°C), relative humidity (RH%), solar radiation (W/m$^2$), wind speed (m/s), and wind direction (°), which were recorded at 15 min intervals and transmitted in real time to the Technical University of Loja (Universidad Técnica Particular de Loja (UTPL)). The experimental burning was executed in all the plots to be burned on the same day between 12 p.m. and 2 p.m., as recommended by Geron and Hays [50], which is characterized as the time when the highest values of solar radiation (W/m$^2$) and temperature (°C) are typically observed, as well as the lowest relative humidity (%).

Before the experimental burning, the fine dead fuel moisture was calculated by applying the procedures outlined in the Interagency Fire Use Module Field Guide [51], which is widely used internationally. Fuel load was calculated by random sampling in the AnP area near the experimental site but outside the plots, using 10 wood squares of 1 m$^2$ each. The squares were randomly distributed, and all herbaceous vegetation was cut with a sickle at ground level. The samples were preserved in paper bags and transported to the laboratory of the UTPL, where they were dried in an oven for 48 h at 60 °C. After dehydrating, the samples were weighed using a Rice Lake TC balance. The fuel load per m$^2$ was expressed in grams of dry matter and the fuel moisture percentage was calculated based on the difference between wet weight and dry weight [52].

Flame height during the experimental fire was measured by placing 4 metal equidistant bars in each plot, with a 5 m interval between each bar. The metal bars were painted with fire-resistant paint and were marked with 5 cm increments to accurately determine the height of the flames in each section. Additionally, photographs were taken during the experiment to confirm the flame height at each bar and calculate the average frame height. The images were carefully coded and analyzed using the UTHSCSA-Imagetool software (The University of Texas Health Science Center, San Antonio, TX, USA) [53].

To measure the soil temperature during the experimental burning, 4 thermocouples were placed per plot at a depth of 5 cm (using EasyLog data logger EL-USB-TC Thermocouples and K-TYPE PROBE 1M5 probe) [54]. The thermocouples were positioned along the centerline of each plot at the same level as the metal bars. The fire propagation speed was determined by using a stopwatch to measure the time it took for the flame front to travel between 2 bars (5 m or 20 m$^2$) and was calculated using the formula provided by Aguirre [55].

Finally, once natural flame extinction occurred, 3 ash samples were collected randomly within each PSP. The samples were carefully stored in labeled plastic Petri dishes. At the laboratory, the ash samples were ground to powder to estimate fire severity using the Munsell chart color method, which has been widely adopted for such purposes [56–58].

### 2.4. Soil Physico-Chemical Properties Analyses

To evaluate the impact of fire on soil physico-chemical properties, samples were collected from burned and unburned plots. One set of samples was taken 24 h after the experimental burn (S1), and the other set was collected 365 days later (S2), following the recommendation of recent studies [59]. This choice was because the most significant changes in soil properties occur in the short and medium term, especially due to the climatic characteristics of the páramo that presents constant precipitation throughout the year and erosive processes can be enhanced in this ecosystem [32,34,48].

Sampling was performed by a single person using the same sampling cylinder, to ensure consistency in the measurements and avoid possible variations that could influence the results. In addition, during samplings S1 and S2, before collecting soil samples, the ash layer was removed according to Santín et al. [60]. In S1, the ash had an average thickness of 3 cm, while in S2, the layer had decreased considerably, presenting a few millimeters of thickness when it was present. For this purpose, the surface of the soil that had been in contact with the ash layer was carefully scraped with a razor to remove any carbonized residue, and then soil samples were collected.

Sampling consisted of collecting 3 soil samples (at the beginning, middle, and end of each PSP), resulting in nine soil subsamples per block, to determine soil bulk density (Bd) and soil water content. We also collected an additional set of nine soil samples at each PSP during collection times S1 and S2, for soil chemical characterization. These nine subsamples were mixed to obtain a composite sample for T1, T2, and the control plots, respectively, for each block. All soil samples were collected at a depth of 10 cm using a standard 283 cm$^3$ metal cylinder (6 cm diameter and 10 cm height) [18]. Each of the samples obtained were placed in plastic bags and labeled appropriately for transport to the laboratory.

In the laboratory, Bd was immediately estimated [61]. Soil water content was determined by weight difference between wet and dry samples (105 °C for 48 h; [61]). Subsequently, an aliquot of the other set of samples intended for the determination of soil texture and chemical properties was dried at room temperature, sieved through a 2 mm mesh, and all visible root debris removed [62]. Soil texture was determined using the hydrometer method [63], while soil pH was measured with a pH meter applying the standard method [61]. Porosity was determined based on the assumption that the density of stones was approximately 2.65 g cm$^{-3}$ [64]. Soil organic carbon (SOC) and soil organic matter (SOM) were determined by the method of Walkley and Black [65,66], for which the sample was placed in an oven at 125 °C for 45 min, after oxidation in a solution of $K_2Cr_2O_7/H_2SO_4$. Soil ammonia nitrogen concentration (SAN; mg/kg) was measured by the colorimetric method [67], while soil available phosphorus (P; mg/kg), potassium (K; meq/100 g), calcium (Ca; meq/100 g), and magnesium (Mg; meq/100 g) were determined by the modified Olsen method [68].

Finally, the equation proposed by Walteros et al. [69] was used to determine soil organic carbon (SOC) content.

$$SOC = OC * Bd * h \tag{1}$$

where SOC is the stock of soil organic carbon (tC ha$^{-1}$ top 10 cm of soil), OC is the total organic carbon concentration (%), Bd is the bulk density (g cm$^{-3}$), and h is the depth at which the sample was taken (cm).

### 2.5. Statistical Analysis

Statistical analyses were performed to evaluate fire behavior and its effect on soil physico-chemical properties. Differences in fire behavior descriptors (flame height, soil temperature, and propagation velocity) were evaluated using a one-way ANOVA (n = 3), with a significance level set at $p < 0.05$ and Tukey's HSD post hoc test ($p < 0.05$). For this purpose, the mean value along different lengths (5 m, 10 m, 15 m, and 20 m) in the experimental plots was considered.

On the other hand, to determine the effect of fire on soil physico-chemical properties at different sampling times (S1, 24 h after burning; S2, one year after burning), a two-way analysis of variance with repeated measures ($p < 0.05$) was used. In addition, to verify the differences between treatments within each sampling time, one-factor ANOVA and Tukey's HSD post hoc test ($p < 0.05$) were performed. Normality and homogeneity of variance assumptions of the data were corroborated by Shapiro—Wilk and Levene tests (Shapiro–Wilk), respectively, before applying parametric tests.

Finally, to evaluate the integrated effects of fire treatments in soil physical properties, such as bulk density (Bd), porosity, and soil moisture, together with changes in chemical

properties (SOM, SOC, pH, SAN, P, K, Ca, and Mg) between the two sample collection times (S1 and S2), a principal component analysis (PCA) was performed. The Pearson correlation coefficient between the soil physico-chemical properties and the scores of PC1 and PC2 was also estimated. All statistical analyses were performed with the statistical program PAST version 3 [70].

## 3. Results

### 3.1. Fire Behavior and Severity of the Burns

The San Lucas_UTPL automatic weather station recorded an average relative humidity of 72.6%, an average air temperature of 15.6 °C, a maximum solar radiation of 1135.1 (W/m$^2$), and an average wind speed of 8.9 m/s (32.0 km/h). It is essential to mention that during the experiment, the wind direction concerning the longitudinal axis of each plot was always in favor of the downhill treatment, with directions mostly from the east to southeast. The absence of precipitation for several days before the experiment indicated good conditions for burning the PSPs with herbaceous páramo vegetation (Table 1).

**Table 1.** Atmospheric conditions during the experimental burning.

| Blocks | Plot | Treatment | Temperature (°C) | Relative Humidity (%) | Wind Speed (m/s) | Wind Direction (°) | Precipitation (mm) | Solar Radiation (W/m$^2$) |
|---|---|---|---|---|---|---|---|---|
| | 1 | T1 = uphill | 14.4 | 77 | 12.1 | 90 | 0 | 1209 |
| 1 | 2 | T2 = downhill | 14.3 | 79 | 8 | 90 | 0 | 492 |
| | 3 | Control plot | 14.5 | 75 | 6.3 | 90 | 0 | 1364 |
| | 4 | T1 = uphill | 15.9 | 72 | 8.9 | 90 | 0 | 1376 |
| 2 | 5 | T2 = downhill | 15.7 | 70 | 8.9 | 157.5 | 0 | 1208 |
| | 6 | Control plot | 16.1 | 70 | 9.4 | 135 | 0 | 1243 |
| | 7 | T1 = uphill | 16.4 | 70 | 8.9 | 135 | 0 | 1155 |
| 3 | 8 | T2 = downhill | 16.5 | 71 | 8 | 135 | 0 | 1164 |
| | 9 | Control plot | 16.2 | 69 | 9.4 | 135 | 0 | 1005 |
| | | Average | 15.6 | 72.6 | 8.9 | 117.5 | 0 | 1135.1 |

According to the danger index of the Interagency Fire Use Module Field Guide [51], the atmospheric conditions during the experimental burn were classified as alerting, with low ignition probability (40%) and a relatively high wind speed of 8.9 m/s (Table 2). Moreover, the combustible moisture content of the vegetation (23.6%) was relatively high, with a wet biomass of 2213.7 g m$^{-2}$ (22.1 t/ha) and a dry biomass of 1790.2 g m$^{-2}$ (17.9 t/ha). It should be noted that soils in the AnPs are generally saturated [71,72], which leads to a classification of moderate fuel moisture content for the herbaceous páramo ecosystem.

**Table 2.** Fine dead fuel moisture according to the Interagency Fire Use Module Field Guide and calculation of fuel load and fuel moisture content.

| | Fine Dead Fuel Moisture (Interagency Fire Use Module Field Guide) | | | | | | Fuel Loading and Humidity | | |
|---|---|---|---|---|---|---|---|---|---|
| Variables | °C | Relative humidity (%) | Shading (%) | Ignition probability (%) | Wind speed (m/s) | Danger index | Wet biomass (kg m$^{-2}$) | Dry biomass (kg m$^{-2}$) | Fuel moisture content (%) |
| | 15.6 ± 0.2 | 72.6 ± 0.9 | 0–10 | 40 | 8.9 ± 0.2 | Alert | 2.21 ± 0.33 | 17.90 ± 0.32 | 23.6 ± 1.5 |

Note: ± is the standard error.

There were no statistically significant differences between the flame heights of the plots burned uphill (T1) and downhill (T2) (*p*-value 0.08). The mean flame height for T1 was 59 cm and for T2 it was 71.6 cm, with the greatest differences observed in the center of

the plots after 10 m (Figure 3a), where T1 reached a flame height of 51.7 cm and T2 reached a flame height of 75.5. Regarding soil temperature, no significant statistical difference was found between T1 and T2 (*p*-value 0.31) (Figure 3b). T1 reached a mean soil temperature of 26.4 °C, while T2 reached a mean soil temperature of 25.8 °C. The greatest differences were registered in the center of the burned plots after 10 m, where T1 had a soil temperature of 28.1 °C and T2 had a soil temperature of 25.5 °C. Soil temperature in the unburned control remained stable at around 13.3 °C. The mean propagation speed for T1 was 0.7 m/minute and for T2 it was 1.0 m/minute, with no significant statistical difference between them (*p*-value 0.06) (Figure 3c,d). However, the propagation speed within the first 5 m was slightly higher in the plots burned downhill (T2: 1.3 m/minute), compared to the T1 plots (T1: 0.7 m/minute), due to the predominate wind direction being from the southeast.

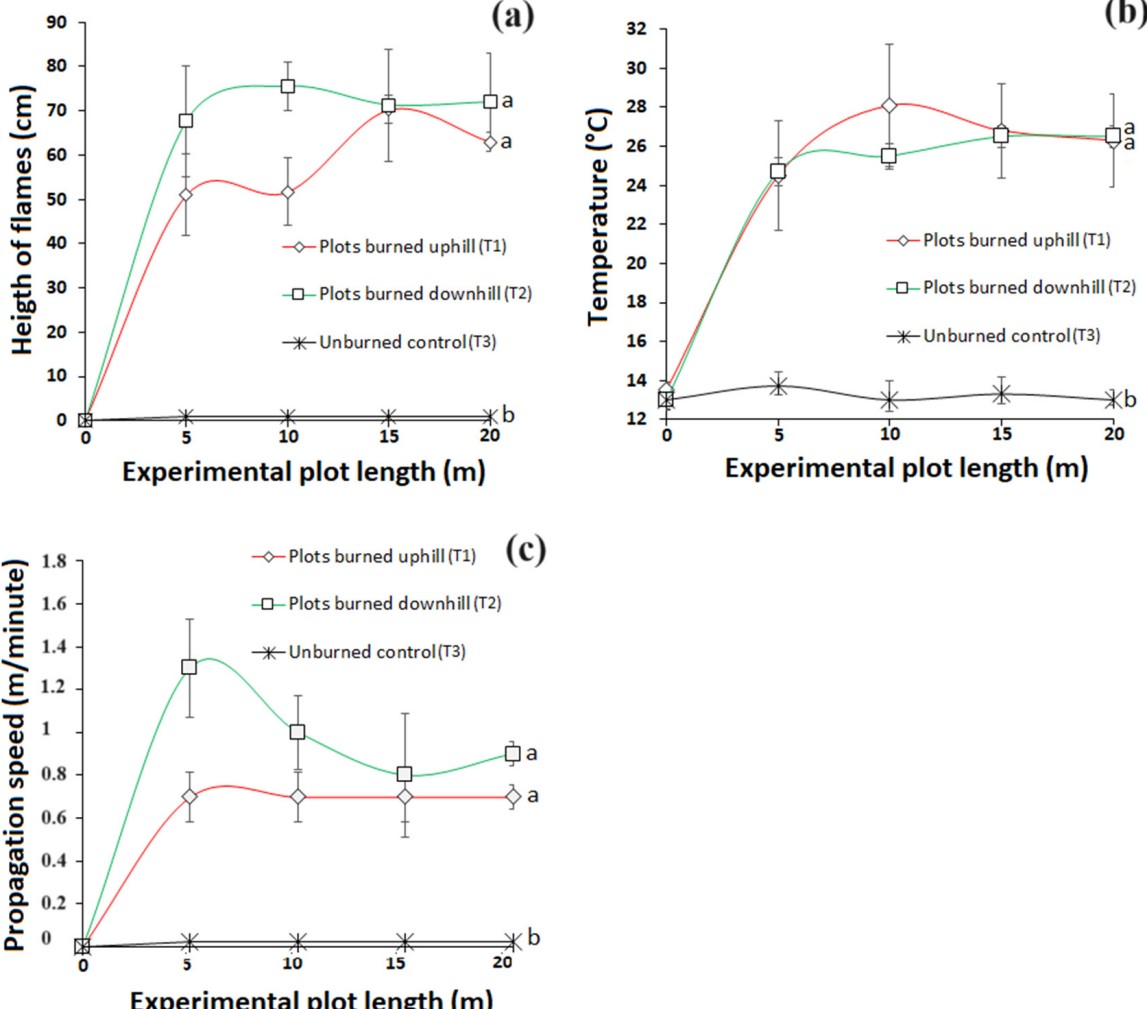

**Figure 3.** Evaluation of fire behavior during experimental burning through an ANOVA (n = 3), with a significance level set at *p* < 0.05. (**a**) Average flame height (cm); (**b**) average soil temperature reached at a depth of 5 cm; (**c**) average fire propagation speed (m/minute). Different letters (a, b) indicate significant difference among burning treatments (*p* ≤ 0.05%, HSD Tukey).

The ash samples from all burned plots had a very dark brown color according to the Munsell color code (=5 yr 2.5/2), which corresponds to low-severity fires [56,57].

### 3.2. Effects of the Experimental Fire on Soil Physico-Chemical Properties

Table 3 and Figure 4 summarize the effects of low-severity fires on the soil physico-chemical properties at a depth of 10 cm in the two sampling times: S1 (4 h after burning) and S2 (one post-fire year). Shortly after burning, Bd was not significantly different between

the burned and unburned plots, including burn directions (uphill slope S1: 0.31 g/cm$^3$; downhill slope S1: 0.30 g/cm$^3$; and control S1: 0.33 g/cm$^3$: *p*-value 0.90). One year after burning, Bd values increased in all plots, indicating a slight soil compaction, but with no significant differences between them (uphill S2: 0.50 g/cm$^3$; downhill S2: 0.46 g/cm$^3$; and control S2: 0.43 g/cm$^3$: *p*-value 0.17) (Table 3; Figure 4a). This increase in Bd over time (S1 vs. S2) was statistically significant (*p*-value 0.02), which was also confirmed by porosity. In addition, porosity showed no significant differences between the burned and control plots in S1 (T1: 88.3%; T2: 88.8%; control: 87.7%: *p*-value 0.75), as well as in S2 (T1: 81.1%; T2: 82.5%; control: 83.87%: *p*-value 0.36). However, a significant decrease over time was observed (*p*-value 0.04; Table 3; Figure 4b). As with Bd and porosity, no significant differences were found for soil moisture in S1 between the burned and control plots (T1: 170.9%; T2: 169.7%; control: 141.2%: *p*-value 0.59), as well as in S2 (T1: 132.8%; T2: 154.0%; control: 126.7%: *p*-value 0.61). However, a sligtdecrease in soil moisture was observed over the different sampling dates (S1 vs. S2; *p*-value 0.05) (Table 3; Figure 4c). In general, as the statistical analyses indicated, fire had no significant effect on the physical soil properties, regardless of the treatment. However, the time factor showed significant changes (S1 vs. S2), in which an increase in Bd and a decrease in porosity, as well as in soil moisture, were observed.

**Table 3.** Effects of low-severity fire on soil physico-chemical properties of herbaceous páramo soil in southern Ecuador. Significant differences were evaluated with repeated measures. Significance level of α = 0.05.

| Variable | Repeated Measures ANOVA S1 and S2 (*p*-Value) | | |
| --- | --- | --- | --- |
| | **Burning (B)** | **Sampling Time (T)** | **B × T** |
| Bd (g cm$^{-3}$) | 0.2844 | 0.0263; S2 > S1 | 0.5487 |
| Porosity (%) | 0.655 | 0.0402; S2 < S1 | 0.3673 |
| Soil water content (%) | 0.6376 | 0.0472; S2 > S1 | 0.2113 |
| SOM (%) | 0.4722 | 0.0701; S2 = S1 | 0.8689 |
| SOC (tC ha$^{-1}$) | 0.9577 | 0.0687; S2 = S1 | 0.6164 |
| pH | 0.0174 | 0.0428; S2 > S1 | 0.1354 |
| SAN (mg/kg) | 0.392 | 0.0384; S2 > S1 | 0.3496 |
| Mg (Meq/100 g) | 0.8267 | 0.015; S2 > S1 | 0.7094 |
| P (mg/kg) | 0.1999 | 0.4701; S2 = S1 | 0.7608 |
| K (Meq/100 g) | 0.1352 | 0.1141; S2 = S1 | 0.006 |
| Ca (Meq/100 g) | 0.2379 | 0.8305; S2 = S1 | 0.2037 |

The fire had no significant effects on SOM or SOC, and this was true for S1 and S2 (Table 3; Figure 4). Both tended to increase, although not significantly, during one year as the values in S2 tended to be higher than those of S1 (Figure 4d,e, respectively).

Although soil pH showed no significant differences between plots in S1 (*p*-value 0.38) and in S2 (*p*-value 0.20), the values were slightly higher in the burned plots. In addition, the pH values decreased with time in all plots, with the control plots always having the lowest values (Table 3, Figure 4f).

SAN showed no significant differences between burning treatments in S1 and in S2 (*p*-value 0.34 and 0.61, respectively) (Table 3). One year after burning (S2), SAN values were significantly higher than those of S1 (Figure 4g), and this was true independently of the fire. Values were as high as 144 mg/kg. No significant differences were found for soil magnesium among the plots in S1 and S2, but it increased significantly over time in all plots (*p*-value 0.02), regardless of the effect of fire. Mg values in S2 were almost four times higher than those measured in S1 in all plots (Table 3; Figure 4h).

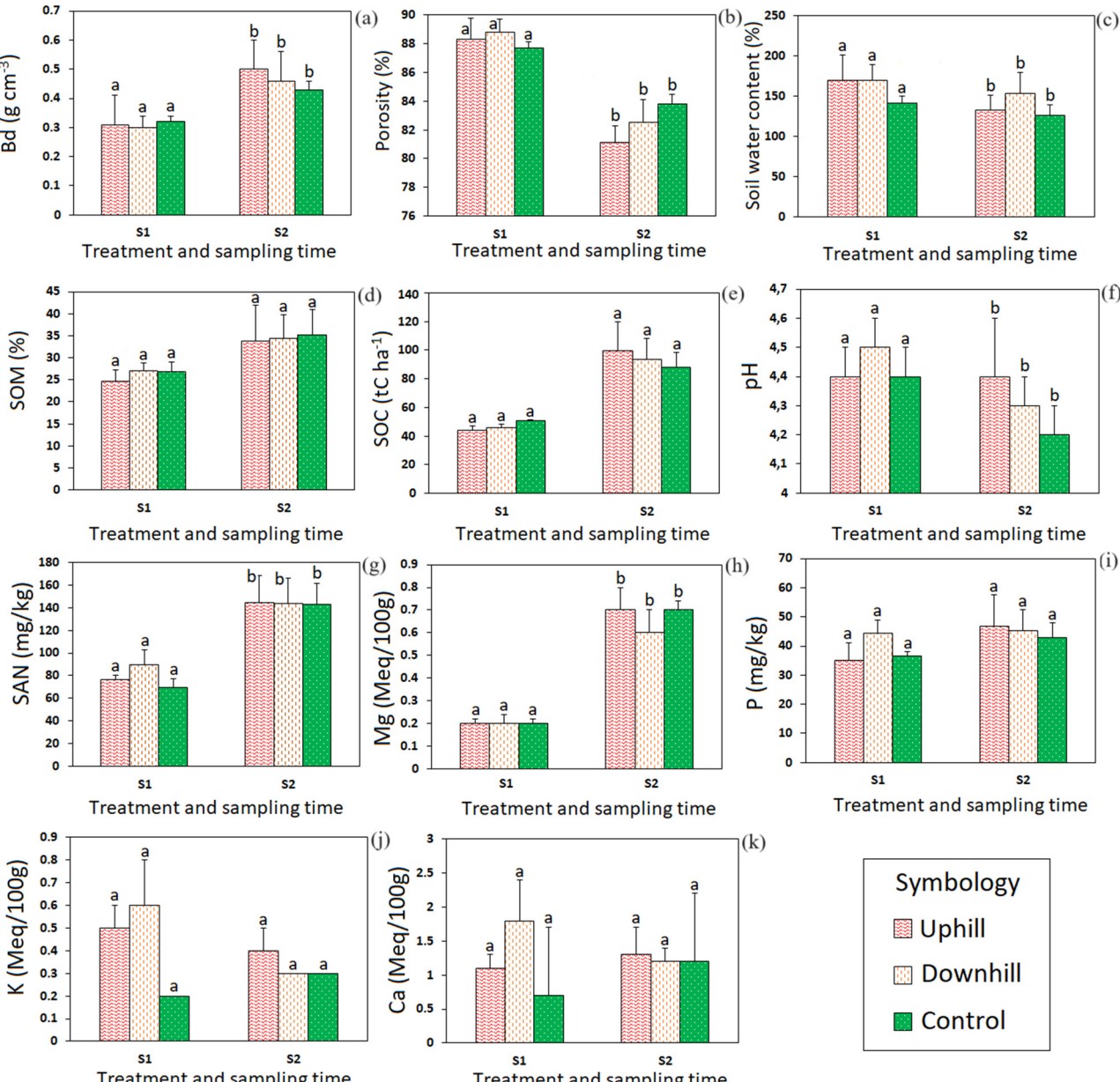

**Figure 4.** Effects of fire treatments at different sampling times on bulk density (Bd) (**a**), porosity (**b**), soil moisture (**c**), soil organic matter (SOM) (**d**), soil organic carbon (SOC) (**e**), soil pH (**f**), soil ammonium nitrogen SAN (**g**), Mg (**h**), P (**i**), K (**j**), and Ca (**k**). Mean values and standard deviation are shown. Different letters indicate significant differences among burning treatments for different sampling times ($p < 0.05$, Tukey HSD).

No significant differences were found for P, either regarding the treatments (*p*-value 0.20) or the time factor (*p*-value 0.47). Only slight increases were observed in all plots one year after burning (Figure 4i). Soil exchangeable potassium (K, Figure 4j) showed an initial increase in the burned plots in S1, followed by a decrease one year later (S2). However, no significant differences in K were observed regarding the treatment (*p*-value 0.13) or the time factor (*p*-value 0.11), but a significant effect was found for the treatment–time interaction (*p*-value 0.01), as the values in the unburned plots did not change over time. Ca (Table 3; Figure 4k) showed an initial slight increase (S1) and a decrease one year later (S2) in the plots burned downhill (T2), while in the T1 and control plots, a slight increase in S2 compared to S1 was observed. However, the differences were not statistically

significant either regarding the treatment (*p*-value 0.24) or the time factor (*p*-value 0.83) or the interaction between them.

### 3.3. Principal Component Analysis (PCA) in the Burned and Unburned Plots

Figure 5 shows the results of the principal component analysis (PCA) of the uphill burning, downhill burning, and control plots after one day (S1) and one year (S1) of the low-severity fire. The first two PCA components, PC1 and PC2, explain 57.3% and 23.4% of the total variance, respectively.

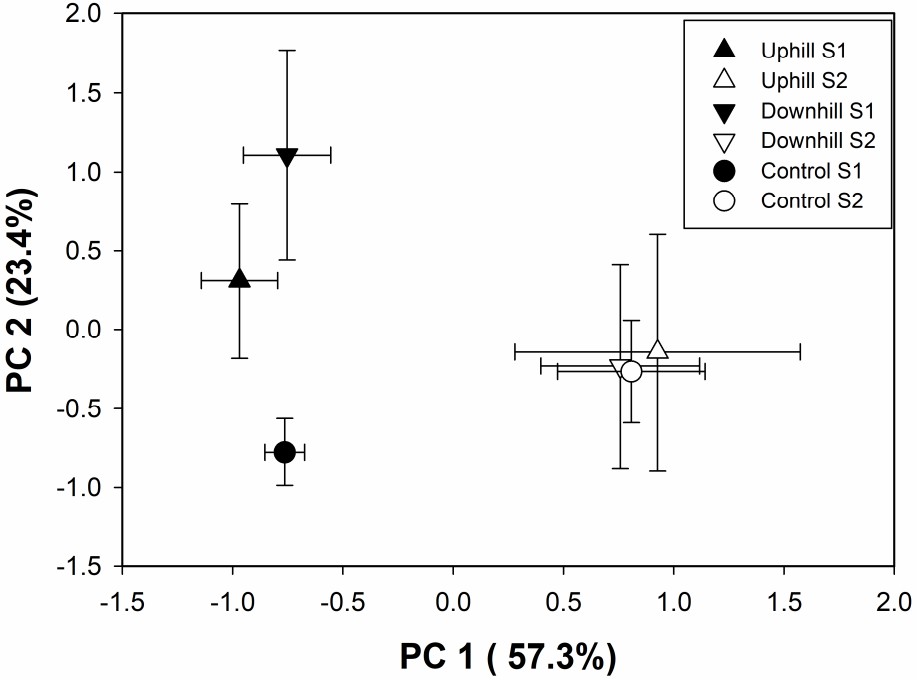

**Figure 5.** Ordination of the different burning treatments for the two sampling times in the space defined by the PC1 and PC2 axis of the PCA analysis carried out with physico-chemical soil properties. Coordinates are the mean of three replicates and bars represent the standard error of the mean.

Bd, SOM, SOC, SAN, and Mg were all significantly and positively correlated with PC1 scores, and porosity and pH were negatively correlated with PC1 scores (Table 4). On the other hand, soil water content, K, and Ca were significantly correlated with PC2 scores (Table 4). PC1 differentiates soils according to their sampling time (Figure 5).

**Table 4.** Pearson product-moment correlation coefficients between soil physico-chemical properties and scores of the PC1 and PC2, which explained, respectively, the 57.3% and 23.4% of the variance in the PCA analysis. Significant coefficients are noted with *.

| Variable | PC1 | PC2 |
|---|---|---|
| Bd (g cm$^{-3}$) | 0.348 * | −0.154 |
| Porosity (%) | −0.349 * | 0.232 |
| Soil water content (%) | −0.101 | 0.498 * |
| SOM (%) | 0.367 * | 0.089 |
| SOC (tC ha$^{-1}$) | 0.386 * | −0.065 |
| pH | −0.319 * | −0.009 |
| SAN (mg/kg) | 0.374 * | 0.165 |
| Mg (Meq/100 g) | 0.367 * | 0.050 |
| P (mg/kg) | 0.262 | 0.298 * |
| K (Meq/100 g) | −0.080 | 0.501 * |
| Ca (Meq/100 g) | 0.105 | 0.541 * |

On the other hand, PC2 differentiates shortly after the fires between the burned and unburned soil, without differences between uphill and downhill burning.

## 4. Discussion

### 4.1. Fire Behavior and Severity during Experimental Burning

Determining the optimal time to carry out the burning of herbaceous vegetation in the páramo presents a challenge due to the frequent rainfall that characterizes this region [73]. Burning requires a rain-free period of at least one week [32], but unfortunately, accurate meteorological information is limited in high mountain areas [62]. Nevertheless, during the execution of the experimental burn, the atmospheric conditions were favorable (VdN phenomenon was recorded), which facilitated the successful ignition of herbaceous vegetation in the páramo (see Table 1) [32]. In this context, the Interagency Fire Use Module Field Guide Hazard Index [51] classified the atmospheric conditions on the day of the burn as alert, with moderate probability of ignition (40%) and a relatively high wind speed (see Table 2). It would have been advisable to have other meteorological conditions to carry out the experimental burns that were within a range of 15–19% relative humidity, 21–31 °C, and a wind speed of 5–11 m/s, based on which the Field Guide danger index would have classified conditions as alarm. Therefore, it is relevant to highlight that this guide was designed for U.S. ecosystems and its applicability may be limited in the unique conditions of the AnP ecosystems. Thus, it is essential to consider its use with caution in this specific region.

Compared to other studies in contrasting ecosystems, flame height in the herbaceous páramo vegetation was lower, mainly due to a lower vegetation height (40 cm) and lower fuel load (22.1 t/ha). For example, Rodríguez-Trejo et al. [74] reported higher flame heights in a 1 m artificial pasture in Chiapas (Mexico) with a fuel load of 6214 t/ha, and under drier climatic conditions. Anderson et al. [75] also found a positive correlation between fuel load and flame height in tropical savannas, indicating that higher fuel loads produce higher flames. Therefore, the flame height and propagation speed observed during the experimental burn are specific to the herbaceous páramo vegetation of the study area and may not be directly comparable to other ecosystems with different vegetation cover and fuel loads.

The severity of the experimental fire could be classified as low, as indicated by the very dark brown color of the ashes. According to Bodí et al. [57], this color indicates incomplete combustion of vegetation, including uncharred organic particles, caused by dehydration and oxidation of iron components. Fires of moderate to high severity are infrequent in AnPs due to the cool and humid weather conditions, which is why soils are generally saturated [76]. However, even low-severity fires could inhibit the regeneration of the AnP ecosystem when heavy rains occur after burning, which enhance erosive processes, causing the loss of the topsoil layer where organic matter and nutrients are concentrated. Therefore, careful management and monitoring of burning activities in AnPs is crucial to minimize the risk of soil degradation and promote ecosystem health. Nevertheless, low-severity fires can also provide benefits to ecosystems, as fire is a natural perturbation of most ecosystems on Earth, and many plant communities have adapted to it [77].

### 4.2. Low-Severity Fire Did Not Affect the Physico-Chemical Soil Properties

The low-severity of the burns did not manifest adverse effects on soil physical and chemical properties in the San Lucas AnP. These findings are consistent with previous research, such as that of Mehdi et al. [78], who reported that low-severity experimental burns did not cause significant alterations in soil properties compared to control sites. These results also align with studies conducted in various global ecosystems, such as those carried out by Agbeshie et al. [79], which corroborate that low-severity burns do not induce deteriorations in soil physical and chemical properties due to incomplete biomass combustion [80].

Low Bd and high porosity values in AnP soils are due to the high SOM concentration, which explains its high water storage capacity [81]. The increased Bd and decreased porosity one year after burning could be due to pore clogging by ash and clay particles, a phenomenon that was previously documented by Woods and Balfour [82]. The presence of these particles in the soil may reduce the water- and air-holding capacity of the pores, resulting in a decrease in porosity over time. This effect is corroborated by Larsen et al. [83], who showed that ash can clog soil macropores, leading to an increase in Bd, reducing soil infiltration rates, and thus increasing surface flow. This effect is especially evident in soils with high porosity, such as AnP soils, in the first 10 months after burning [82]. Later, however, soil porosity recovers, especially in regions with high amounts of previous precipitation [57]. Likewise, Bd values did not reach the level of soil compaction, which could inhibit germination [84], as values higher than 1.45 g cm$^{-3}$ indicate this condition.

Consequently, low-severity burns in AnP do not negatively affect soil physical properties, as the destruction of soil aggregates by low-severity fires is not to be expected [85]. Furthermore, low-severity fires do not compromise the hydrological properties of AnP soils, even though fire usually reduces soil moisture by evaporation [86]. Although soil moisture decreased over time in the study area, this cannot be related to the experimental fire, as it depends on the fire temperature and soil heating, which were minimal in this AnP study, and due to climatic conditions, which show high annual and interannual variations in the Andes of southern Ecuador [87].

In addition, the low-severity experimental burns did not cause negative effects on the chemical properties of the studied AnP soils, as reported in recent research [79]. For example, the observed slight increase in SOM in all plots one year after burning (S2: upslope 33.7%; downslope 43.4%; and control 35.2%) could be due to the fact that the fencing of the study area led to increased litter deposition and facilitated natural succession, as the burned area was protected from grazing by domestic and wild animals (browsing of vegetation was prevented), which also contributed to SOM accumulation over time [14]. However, these values are relatively lower than those reported by Patiño et al. [88], who, analyzing 33 scientific papers in South American AnPs, determined an average of 43% SOM in the soil. Therefore, to better understand these results, it is recommended to continue monitoring soil organic matter content over time and to consider other factors such as soil erosion that could be influencing the observed changes.

On the other hand, the cold and humid climate of the AnP also favors SOM accumulation [2], and this process is enhanced by the formation of organometallic complexes, known as an association between organic matter and soil mineral particles ($Al_{3+}$; $Fe_{3+}$) [89]. This increase in SOM could also be favored by low fire severity, the slow decomposition of burned biomass, and ash addition [79]. However, it should be noted that a high SOM content has benefits for the edaphic ecology of this páramo since it is known that SOM allows the storage of a large amount of water, improving infiltration rates and hydraulic conductivity [90], which improves soil structure and leads to greater granularity and increased root development [91]. These edaphic conditions can especially favor the growth of native páramo species that are adapted to the adverse climatic conditions of these high-altitude areas, such as puya (*Puya aequatorialis*), páramo straw (*Calamagrostis intermedia*), páramo flower (*Oritrophium crocifolium*), naurapo (*Myrteola nummularia*), and joyapas (*Disterigma rimbachii*) [40,41].

Moreover, the climate in the studied AnP and the high water storage capacity of the soils lead to slow rates of the decomposition and accumulation of organic matter, which makes this ecosystem an important carbon sink [6]. This is corroborated by Medina and Mena [92], who reported that AnPs store six times more carbon than tropical forests, which is illustrated by the high SOC values calculated in this study at a soil depth of 10 cm (average S2: 93.7 tC ha$^{-1}$; Table 3). These values are similar to the SOC values published by Santín and Vidal [60] for different soil types in a shrubby páramo area in Podocarpus National Park (PNP) in southern Ecuador (91.52 tC ha$^{-1}$ for inceptisol soils and 68.37 tC ha$^{-1}$ for entisol soils). However, the type of management and geographic location of the AnP also

affect SOC. Quiroz Dahik et al. [93] only reported values of around 1.5 tC ha$^{-1}$ in pastoral páramos (with extensive and intensive grazing) near the equator, while Cargua et al. [94] reported values of up to 277.81 tC ha$^{-1}$ in páramos located on volcanoes due to frequent ash expulsion.

Another parameter that was affected by low-severity burns was pH. During the two sample collection times (S1 and S2), pH values in all PSPs ranged between 4.1 and 4.5, indicating suboptimal acidity for agricultural purposes, as the optimal soil pH for agriculture is between 5.0 and 8.0 [95]. According to Neina [96], low pH can negatively affect soil biogeochemistry and limit the availability of essential nutrients for plant growth. However, native, and endemic plant species in AnPs, which include native vascular plants, grasses, bryophytes, and lichens, may be adapted to these acidic soils. To elucidate this question, some researchers used acid phosphatase enzyme activity to determine soil quality/health in the tropical Andes [18]. For example, Turner [97] found that in tropical soils, acid phosphatase activity increases at a pH optimum of around 4.0, which is close to the values observed in this study. This may further explain the relatively high values of available phosphorus (P) in páramo soil (Figure 4i) and highlights the importance of soil pH in páramo ecology. Therefore, further investigation of the role of pH in this unique ecosystem is needed.

Similarly, low-severity burning did not have a negative effect on soil nutrients, as previous studies have shown [35]. In addition, it is important to note that low-severity burning did not result in nitrogen losses, as would sometimes be expected, but rather transformed it into a more plant-accessible form, thanks to suitable soil temperature and pH conditions [98].

As with SAN, magnesium (Mg; Figure 4h) also showed significant increases one year after burning (S2), even though it was expected that values would increase immediately after burning (S1) due to the combustion of organic materials, and decrease subsequently due to ash erosion, leaching, and plant uptake [99]. However, the slow decomposition process at these high altitudes and the increase in SOM led to an accumulation of Mg in the soils, which was not only observed in the burned plots, but also in the unburned control plots, probably due to the distribution of Mg over wider areas by surface and subsurface flows and by the protection of the study area with wooden posts and barbed wire.

To a lesser extent, available phosphorus (P; Figure 4i), which is the second most limiting plant nutrient after nitrogen, showed a slightincrease during S2. P becomes available through the transformation of organic P into organic matter during fire (i.e., mineralization), which increases P in the upper soil horizons [100]. The high P concentration observed during S2 (40.0 mg kg$^{-1}$) could be attributed to the low soil pH and the presence of mycorrhizal fungi that release organic acids, which promote weathering and the decomposition of organic matter, leading to increased soil P [101].

The soil exchangeable potassium (K; Figure 4j) showed the expected pattern of an initial increase immediately after burning (S1), followed by a decrease one year later (S2) due to leaching and plant uptake [102]. Interestingly, calcium (Ca; Figure 4k) showed an increase in the burned plots in S1, but the decrease in S2 was only observed in the plots burned downhill (T2), while the values in the plots burned uphill (T1) stayed stable and the values in the control plot increased. This could confirms the nutrient transport and distribution by surface and subsurface flows in this precipitous terrain (see Figure 2).

In summary, this study demonstrates that low-severity burns do not have negative effects on soil physico-chemical properties in the herbaceous páramo ecosystems of southern Ecuador. In addition, one year after burning (S2), an increase in SOM, SOC, SAN, and Mg contents was observed. Likewise, the available phosphorus (P) content increased due to the low soil pH (see Figure 5, Table 4), which increases acid phosphatase activity [97]. These results are consistent with recent studies, which indicate that low-severity fires increase nutrient availability and favor post-fire natural regeneration as well as plant community growth [35,100].

However, our study cannot verify any increase or decrease in nutrient availability that can be confidently attributed to fire. Therefore, further research on burning practices in AnP, including fire severity and frequency, is needed to better understand their impact on post-fire soil and plant development [103].

Looking to the future, it is important to control and reduce anthropogenic pressures on AnPs due to their capacities as carbon sinks and for water storage. Conversely, an intensive conversion to agricultural land and cattle ranching should be avoided, as well as avoiding the establishment of pine plantations (*Pinus patula* and *Pinus radiata*) and large-scale mining as occurs in other AnPs in Ecuador [6]. According to Avellaneda-Torres et al. [104], all of these activities have negative impacts on water supply and nutrient availability, which subsequently alter natural succession and intensify soil erosion processes, which could lead to the loss of the biological structure of the AnP when the topsoil is completely lost [22]. It is therefore crucial to adopt sustainable management practices that consider the fragility of the ecosystem and the services it provides to humans and wildlife (water regulation, carbon sink), in addition to biodiversity conservation. Such practices should give priority to the conservation and restoration of AnPs, including alternative income for the livelihoods of local people, such as sustainable tourism in this unique ecosystem.

## 5. Conclusions

This study analyzed the impact of low-severity fire on soil physico-chemical properties in a herbaceous páramo in southern Ecuador. The results indicate that the optimal period for burning is during the Veranillo del Niño (VdN) phenomenon, which is characterized by favorable climatic conditions involving a decrease in relative humidity, increased temperature, and solar radiation, and, especially, the absence of precipitation. The study concluded that low-severity burns have no significant effects on soil physico-chemical properties, in terms of soil organic matter and nutrient availability, even one year after burning. However, long-term monitoring is necessary to analyze soil erosion processes and succession dynamics, as well as nutrient availability due to leaching over time. Conserving this natural ecosystem and protecting it from anthropogenic activities, such as intensive agriculture and cattle ranching, is essential to ensure ecosystem services such as water supply for the local population and the reduction in food risks in the valleys. In addition, AnPs are important carbon sinks that absorb significant amounts of the anthropogenic greenhouse gas emissions [105], which is essential for buffering the effects of ongoing climate change. Finally, it is crucial to implement comprehensive fire management policies to ensure sustainable management of AnPs.

**Author Contributions:** Conceptualization, V.C.-P. and A.O.; methodology, V.C.-P., A.O., M.B.H., L.J.Á. and R.G.-R.; software, V.C.-P. and M.B.H.; validation, M.B.H., Á.B., A.F., F.L.R. and R.G.-R.; investigation, V.C.-P., M.B.H., L.J.Á. and R.G.-R.; resources, V.C.-P. and A.O.; data curation, V.C.-P.; writing—original draft preparation, V.C.-P., A.F. and A.O.; writing—review and editing, V.C.-P., M.B.H., L.J.Á., F.LR., Á.B. and R.G.-R.; visualization, V.C.-P.; supervision, R.G.-R.; project administration, V.C.-P. and A.O.; funding acquisition, V.C.-P. and F.L.R. All authors have read and agreed to the published version of the manuscript.

**Funding:** This research was carried out with the technical support of the trilateral cooperation project "Strengthening of Technical and Institutional Capacities in Ecuador for Integrated Fire Management in Conservation Areas" implemented by the German Cooperation for Sustainable Development through the Deutsche Gesellschaft für Internationale Zusammenarbeit (GIZ) GmbH, the Amazonía Sin Fuego Ecuador Program of the Ministry of Environment and Water of Ecuador, by the Brazilian Cooperation Agency (ABC), the National Center for Prevention and Combat of WildFires—Prevfogo of the Brazilian Institute of Environment and Renewable Natural Resources (Ibama of Brazil), and the National System of Conservation Areas (SINAC) of Costa Rica. The funding code is PROY_CONS_CCBIO_2020_2751.

**Institutional Review Board Statement:** Not applicable.

**Informed Consent Statement:** Not applicable.

**Data Availability Statement:** Data are contained within the article.

**Acknowledgments:** Our thanks go to the Universidad Técnica Particular de Loja and GIZ_Ecuador for funding this research (PROY_CONS_CCBIO_2020_2751).

**Conflicts of Interest:** The authors declare no conflict of interest.

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
