# Peer review of "Effects of Low-Severity Fire on Soil Physico-Chemical Properties in an Andean Páramo in Southern Ecuador"

_fire, doi:10.3390/fire6120447_

Round 1
Reviewer 1 Report
Comments and Suggestions for Authors
I reviewed manuscript ID: 2679619 submitted to MDPI journal FIRE entitled “Low-severity fire did not alter the soil physiochemical properties in an Andean Paramo in southern Ecuador” by V. Carrion-Paladines, A. Fries, M.B. Hinojosa, A. Ona, and four others.
The overall objective of the study was to evaluate the short and long-term effects of two methods of burning herbaceous fuels on soil properties in the Andean Paramo region of southern Ecuador. The study addressed relevant questions associated with the effects of fire on conservation and management of paramos ecosystems. Overall, I found the manuscript was well organized, clearly written and provides an adequate description of the study design and results. However, I have several comments and questions after reviewing the manuscript.
The introduction provides an adequate review of literature and a description of why the study is relevant. The methods and results were well written and provided most of the necessary information I was looking for. The study design includes a no-burn, control treatment. Although not stated, I presume another phase of the study will include vegetative response to the fire treatments, which will increase comparisons with the control treatment. The discussion is adequate and includes many references comparing findings of this study with other research.
The authors appear to have used excessive abbreviations, some of which are used only once: HMF (Line 103); FSFM (Line 174), and perhaps others.
Throughout the manuscript, mean values are provided, but without a measure of variation, such as standard deviation or standard error. Lack of an indication of variation is unusual in scientific journal papers and would be appropriate here.
My specific questions are:
Line 114: is “reedgras” spelled correctly?
Line 142: Figure 2. Good photo that helps me better visualize the environment of the study site. However, it appears that the 3 treatments were assigned uniformly, rather than randomly within each of the 3 blocks. It seems unusual that this consistent arrangement of treatments would have resulted from random assignments, which is necessary for appropriate experimental design. If treatments were assigned randomly, that fact should be stated in Line 139.
Line 147: Adding supplementary fuel to the burning plots does not seem appropriate to me. Was more fuel also added to the control plots? At the very least, the amount of additional fuel should be provided and explanation of why it was not added to the control plots. Reference [43] is provided in this sentence, but my review of that publication revealed nothing about the subject of this sentence, which is supplementary fuel. Is [43] relevant in this sentence?
Line 198: Perhaps I missed seeing it elsewhere, but were the 2 burning treatments in the 3 blocks made the same day? Also, is there any record of the last time the study area was burned, which addresses the question of years of fuel accumulation present for these burning treatments.
Line 227: Abbreviation Bd has been defined at Line: 220.
Line 337: Table 3: I do not understand the presentation of information in this table. There is much “white-space” for each variable, but only one line of data presenting results of the ANOVA. Because only one significant difference was found (for K) that could be presented in the text and Table 3 eliminated. However, I may not be interpreting the purpose of providing this table or perhaps something is missing.
Line 369: Figure 4: Be consistent with acronyms and their definitions. For example, “Bd” is used for the y-axis of panel (a) and should be defined in the figure caption as bulk density (Bd) (a), and soil organic matter (SOM) (d). Soil ammonium nitrogen SAN (g) is done properly, but not for other variables.
Line 342: Should word “not” be “no”?
Line 420: Consider replacing words “due to” with “as indicated by”. As the sentence now reads, it implies low fire severity was because of very dark brown color, when the reverse was the situation as I understand it, that is the dark brown color indicated a low severity fire.
Line 696: In reference [43], word “years” should be “year.”
Author Response
Rewiever 1
Comments and Suggestions for Authors
I reviewed manuscript ID: 2679619 submitted to MDPI journal FIRE entitled “Low-severity fire did not alter the soil physiochemical properties in an Andean Paramo in southern Ecuador” by V. Carrion-Paladines, A. Fries, M.B. Hinojosa, A. Ona, and four others.
The overall objective of the study was to evaluate the short and long-term effects of two methods of burning herbaceous fuels on soil properties in the Andean Paramo region of southern Ecuador. The study addressed relevant questions associated with the effects of fire on conservation and management of paramos ecosystems. Overall, I found the manuscript was well organized, clearly written and provides an adequate description of the study design and results. However, I have several comments and questions after reviewing the manuscript.
The introduction provides an adequate review of literature and a description of why the study is relevant. The methods and results were well written and provided most of the necessary information I was looking for. The study design includes a no-burn, control treatment. Although not stated, I presume another phase of the study will include vegetative response to the fire treatments, which will increase comparisons with the control treatment. The discussion is adequate and includes many references comparing findings of this study with other research.
Vinicio
The authors appear to have used excessive abbreviations, some of which are used only once: HMF (Line 103); FSFM (Line 174), and perhaps others.
Thank you for your comment. We agree with the reviewer's suggestion. Accordingly, we have removed the abbreviations in lines 103 and 186 since they are not repeated elsewhere in the text. We thank the reviewer for this observation.
Throughout the manuscript, mean values are provided, but without a measure of variation, such as standard deviation or standard error. Lack of an indication of variation is unusual in scientific journal papers and would be appropriate here.
Vinicio
My specific questions are:
Line 114: is “reedgras” spelled correctly?
We have replaced "redgras" with "paramo straw (Line 114)," as this accurately represents páramo straw, as reported by Torres, M. C., Naranjo, E., & Fierro, V. (2023). Challenges Facing Andean Communities in the Protection of the Páramo in the Central Highlands of Ecuador. Sustainability, 15(15), 11980. We appreciate the reviewer for pointing this out.
Line 142: Figure 2. Good photo that helps me better visualize the environment of the study site. However, it appears that the 3 treatments were assigned uniformly, rather than randomly within each of the 3 blocks. It seems unusual that this consistent arrangement of treatments would have resulted from random assignments, which is necessary for appropriate experimental design. If treatments were assigned randomly, that fact should be stated in Line 139.
We thank the reviewer for this comment on what we erroneously called "blocks," as their effect was not tested. We were not interested in its effect because of the high comparability between plots, suggested by their very similar edaphic properties and plant community before the fire. We have tried to clarify this by not using this term and considering the study plots as independent experimental units (Line 137 - 138) (we have also adjusted Figure 2; line 146).
In this context, collecting unbiased monitoring data on fire effects is often problematic. Samples collected to assess wildfire effects are often "pseudoreplicated" because it is impossible to reproduce the disturbing event. In this sense, experimental studies, such as the one presented here, are invaluable for understanding the effects of fires because they make it possible to reproduce the disturbance event. Thus, it is widely assumed that separate fires in each system can be treated as replicates and avoid many analytical problems (van Mantgem et al., 2001).
Reference:
van Mantgem, P., Schwartz, M., Keifer, M., 2001. Monitoring fire effects for managed burns and wildfires: conning to terms with pseudoreplication. Nat. Areas J. 21, 266-273.
Line 147: Adding supplementary fuel to the burning plots does not seem appropriate to me. Was more fuel also added to the control plots? At the very least, the amount of additional fuel should be provided and explanation of why it was not added to the control plots. Reference [43] is provided in this sentence, but my review of that publication revealed nothing about the subject of this sentence, which is supplementary fuel. Is [43] relevant in this sentence?
Thank you for your comment. We have clarified the need to add supplemental fuel in the combustion plots because in the páramo studied the Saraguro Indians for their traditional burning build mounds with the dry vegetation and then proceed to burn them. The purpose was to simulate what the Indians do in our ignition plots, so we have added the respective bibliographic citation [32]. Also, although the fuel was not weighed, we tried to distribute it evenly as explained in the text (Lines 152 to 157).
On the other hand, we did not include fuel in the control plots, since we used them as "areas not affected by fire", as has been proposed by other researchers [43] (Line 158).
Line 198: Perhaps I missed seeing it elsewhere, but were the 2 burning treatments in the 3 blocks made the same day? Also, is there any record of the last time the study area was burned, which addresses the question of years of fuel accumulation present for these burning treatments.
Thank you for your comment. We have added in the text that the experimental burns were conducted on the same day (lines 181 to 182) and have written about the reviewer's concern about the question of fuel accumulation years in the study area (lines 138 to 142). Unfortunately, we do not have a record with a precise date of the last time the study area was burned. The lack of this record is due to the absence of specific historical documentation of past fires in this páramo. To address this issue (fuel accumulation), we had the valuable collaboration of the indigenous Saraguros (users of this ecosystem). Their knowledge and experience in fire management in this area allowed us to select the appropriate study site for our research. However, the lack of accurate records makes it difficult to provide an exact figure of the years of fuel accumulation prior to our burning treatments.
Line 227: Abbreviation Bd has been defined at Line: 220.
Thank you for your comment. It has been corrected (it is now on line 239).
Line 337: Table 3: I do not understand the presentation of information in this table. There is much “white-space” for each variable, but only one line of data presenting results of the ANOVA. Because only one significant difference was found (for K) that could be presented in the text and Table 3 eliminated. However, I may not be interpreting the purpose of providing this table or perhaps something is missing.
We agree with the review comments. In the revised version of the manuscript, Table 3 has been edited to avoid "white space" and make the table clearer (lines 345 to 349).
Line 369: Figure 4: Be consistent with acronyms and their definitions. For example, “Bd” is used for the y-axis of panel (a) and should be defined in the figure caption as bulk density (Bd) (a), and soil organic matter (SOM) (d). Soil ammonium nitrogen SAN (g) is done properly, but not for other variables.
Thanks for noting the inconsistency of the use of definition and acronyms. It has been corrected in the main text (Line 378).
Line 342: Should word “not” be “no”?
Thank you corrected (line 351).
Line 420: Consider replacing words “due to” with “as indicated by”. As the sentence now reads, it implies low fire severity was because of very dark brown color, when the reverse was the situation as I understand it, that is the dark brown color indicated a low severity fire.
Thank you for your comment, we have replaced "due to" by "as indicated by" (line 430), now it is clearer.
Line 696: In reference [43], word “years” should be “year.”
Thanks for the comment, we have corrected the word "years" to "year" (Line 705).

Reviewer 2 Report
Comments and Suggestions for Authors
The article “Low-Severity Fire Did Not Alter the Soil Physicochemical Properties in an Andean Páramo in Southern Ecuador” discusses the effects of experimental fires on soils of unique ecosystems in Andean páramo. The relevance of this study for Andean Páramo region is beyond doubt. Novelty is well-defined in Introduction. The study is presented in a well-structured manner. The results are described and well discussed. The conclusions are supported by the results. I have following comments to make about this work:
1. I would recommend transforming the title of the article, as the current title does not promote much interest in further reading, because this title makes everything clear. This is at the authors and editor's discretion.
2. Material and methods: 1) It is not clear why there are 3 blocks in the experiment. What is the difference between the blocks? This information should be added. 2) Why ash layer was removed before sampling? 3) Why the method of Walkley and Black was chosen to determine soil organic carbon content? It is known that this method is not suitable for samples with a concentration of more than 15% organic carbon, because in this case the completeness of organic matter oxidation is not achieved.
3. Results: 1) Table 2 - No standard deviation values are given for parameters such as wet biomass, dry biomass and fuel moisture content, despite the fact that 10 samples were taken. These parameters are highly variable; 2) Is there any difference in ash production between plots burned uphill and downhill? If this has been evaluated, please, provide this information.
4. It is better to choose a uniform style of writing the term "physicochemical" or "physic-chemical".
5. Lines 32, 87, 88, 92, 337, 462, 546, 573 - there should be low-severity…
6. Line 235 - extra letter “e”.
7. Line 283 - there should be t/ha.
8. Figure 4. - missing word in “Mean values and are shown”. “Different letters indicate significant differences among burning for different sampling times (p < 0.05, Tukey HSD)” - which letters?
9. Line 412 - Rodríguez-Trejo et al.
10. Line 445 - extra bracket.
11. Line 489 - by Medina and Mena.
12. References - errors in the names, surnames of authors (# 69, 80, 83, 85, 88).
Author Response
Rewiever 2
Comments and Suggestions for Authors
The article “Low-Severity Fire Did Not Alter the Soil Physicochemical Properties in an Andean Páramo in Southern Ecuador” discusses the effects of experimental fires on soils of unique ecosystems in Andean páramo. The relevance of this study for Andean Páramo region is beyond doubt. Novelty is well-defined in Introduction. The study is presented in a well-structured manner. The results are described and well discussed. The conclusions are supported by the results. I have following comments to make about this work:
- I would recommend transforming the title of the article, as the current title does not promote much interest in further reading, because this title makes everything clear. This is at the authors and editor's discretion.
Thank you for your recommendation and we agree with the reviewer. We have improved the title to: Effects of low-severity fire on soil physico-chemical properties in an Andean paramo southern Ecuador.
- Material and methods: 1)It is not clear why there are 3 blocks in the experiment. What is the difference between the blocks? This information should be added.
We thank the reviewer for his comment on what we erroneously called "blocks", as their effect was not tested. We were not interested in their effect because of the high comparability between plots, suggested by their very similar edaphic properties and pre-fire plant community. We have tried to clarify this by not using this term and considering the study plots as independent experimental units.
2) Why ash layer was removed before sampling?
We thank the reviewer for his comment. The removal of the ash layer prior to sampling was carried out to prevent the possible influence of ash on subsequent analyses of soil physicochemical properties. Although it is true that ash could affect both short and long term, our study focused exclusively on the physicochemical properties of the soil and not on the characteristics of the ash or its combination. This method follows a commonly accepted practice in the scientific literature.
3) Why the method of Walkley and Black was chosen to determine soil organic carbon content? It is known that this method is not suitable for samples with a concentration of more than 15% organic carbon, because in this case the completeness of organic matter oxidation is not achieved.
Thank you for your comment. It is true that the Walkey and Black method is not fully suitable for samples with a SOC concentration higher than 15%. However, as far as we are concerned, there are fully adequate methods for direct determination of SOC in soils with SOC concentration above 15-20 %. Our SOC data ranged from (13.4% to 24.8%), which is very close to the SOC threshold of 15 %, and therefore we believe that we are not incurring significant inaccuracies. However, the upper standard of the calibration curve (with an r2 of 0.0993) was set at 25 % SOC. In addition, many researchers have used this method on paramos soils. Some examples are given below:
Echeverría, M., Mur, R. J., Lindao, V., Erazo, N., Logroño, W., & Córdova, R. (2018, August). Quantification of organic carbon stored in the soil in the paramo of Igualata, Chimborazo province-Ecuador. In AIP Conference Proceedings (Vol. 2003, No. 1). AIP Publishing.
Benavides, I. F., Solarte, M. E., Pabón, V., Ordóñez, A., Beltrán, E., Rosero, S., & Torres, C. (2018). The variation of infiltration rates and physical-chemical soil properties across a land cover and land use gradient in a Paramo of southwestern Colombia. Journal of Soil and Water Conservation, 73(4), 400-410.
Avellaneda-Torres, L. M., Sicard, T. E. L., & Rojas, E. T. (2018). Impact of potato cultivation and cattle farming on physicochemical parameters and enzymatic activities of Neotropical high Andean Páramo ecosystem soils. Science of the Total Environment, 631, 1600-1610.
- Results:
1) Table 2 - No standard deviation values are given for parameters such as wet biomass, dry biomass and fuel moisture content, despite the fact that 10 samples were taken. These parameters are highly variable;
Thank you for your comment. We agree with the reviewer that these parameters vary considerably, especially in the herbaceous páramo, which presents a mixture of páramo straw (Calamagrostis effusa), bryophytes and small lichens, along with the presence of achupallas (Puya aequatorialis), the latter containing more biomass. This generates a great variability in the samples taken at random in each square meter, since the presence of achupallas was only found in four of the samples. For this reason, we have included the standard error for wet biomass, dry biomass, and fuel moisture content to communicate the reliability of the estimates based on these samples, and to help readers interpret the robustness of the results presented.
2) Is there any difference in ash production between plots burned uphill and downhill? If this has been evaluated, please, provide this information.
Thank you for your comment. In this study, we did not perform a quantitative assessment of the amount of ash produced in the burned plots, so we do not have specific calculations on this aspect. Instead, we focused on determining fire severity using ash color, following the methodology proposed by Pereira et al. (2014) and Bodí et al. (2014).
- It is better to choose a uniform style of writing the term "physicochemical" or "physic-chemical".
Thank you for your comment. We have corrected physicochemical to physico-chemical throughout the document. The reviewer is thanked for this observation, now the text is clearer.
- Lines 32, 87, 88, 92, 337, 462, 546, 573 - there should be low-severity…
Thank you for your comment. It has been corrected in the text.
- Line 235 - extra letter “e”.
Thank you for your comment, it has been corrected.
- Line 283 - there should be t/ha.
Thank you for your comment. Corrected as requested by the reviewer (Now line 297).
- Figure 4. - missing word in “Mean values and are shown”. “Different letters indicate significant differences among burning for different sampling times (p < 0.05, Tukey HSD)” - which letters?
Thank you for your comment. It has been corrected in the text, as it follows: “Mean values and standar deviation are shown. Different letters indicate significant differences among burning treatments for different sampling times (p < 0.05, Tukey HSD).”
- Line 412 - Rodríguez-Trejo et al.
Thank you for your comment. Corrected as requested by the reviewer (now line 424).
- Line 445 - extra bracket.
Thank you for your comment. Corrected as requested by the reviewer (now line 457).
- Line 489 - by Medina and Mena.
Thank you for your comment. Corrected as requested by the reviewer (now line 501).
- References- errors in the names, surnames of authors (# 69, 80, 83, 85, 88).
Thank you for pointing out these errors in the references. We have corrected the names and surnames of the authors in references 69, 80, 83, 85 and 88. We appreciate your observation and are grateful for your attention.

Round 2
Reviewer 2 Report
Comments and Suggestions for Authors
All comments on the article have been taken into account, and the authors have given comprehensive answers.
Title. I suggest a variant below, but I don't insist. This is at the authors and editor's discretion.
Does low-intensity fire alter soil physico-chemical properties in an Andean Paramo southern Ecuador?
In Table 2, it is better to add a note that +- is the standart error